# A Constitutive Model Study of Chemical Corrosion Sandstone Based on Support Vector Machine and Artificial Bee Colony Algorithm

**Yun Lin [1,2,3], Chong Li [1], Keping Zhou [1], Zhenghai Guo [1] and Chuanwei Zang [2,*]**

[1] School of Resources and Safety Engineering, Central South University, Changsha 410083, China; yunlin617@163.com (Y.L.); lichong2076@163.com (C.L.); kpzhou@vip.163.com (K.Z.); 225511067@csu.edu.cn (Z.G.)

[2] Key Laboratory of Mining Disaster Prevention and Control, Qingdao 266590, China

[3] State Key Laboratory for Geomechanics and Deep Underground Engineering, China University of Mining & Technology, Xuzhou 221000, China

[*] Correspondence: chuanweizang@163.com; Tel.: +86-532-8605-8083

**Abstract:** The mechanical characteristics of rock are greatly influenced by hydrochemical corrosion. The chemical corrosion impact and deformation properties of the meso-pore structure of rock under the action of different hydrochemical solutions for the stability evaluation of rock mass engineering are of high theoretical relevance and applied value. Based on actual data, a support vector machine (SVM) rock constitutive model based on artificial bee colony algorithm (ABC) optimization is constructed in this article. The impact of porosity (chemical deterioration), confining pressure, and other aspects is thoroughly examined. It is used to mimic the triaxial mechanical behavior of rock under various hydration conditions, with high nonlinear prediction ability. Simultaneously, the statistical damage constitutive model and the ABC-SVM constitutive model are used to forecast the sample's stress–strain curve and compare it to the experimental data. The two models' correlation coefficients ($R^2$), root mean square error (RMSE), and mean absolute percentage error (MAPE) are computed and examined. The correlation coefficient between the ABC-SVM constitutive model calculation results and the experimental results is found to be larger ($R^2 = 0.998$), and the error is smaller (RMSE = 0.7730, MAPE = 1.51), indicating that it has better prediction performance on the conventional triaxial constitutive relationship of rock. It is a highly promising new way of describing the rock's constitutive connection.

**Keywords:** hydrochemical corrosion; constitutive relationship of sandstone; machine learning; artificial bee colony algorithm (ABC); support vector machine (SVM)

## 1. Introduction

The change in the relationship between stress and the related strain of rock under external force is referred to as the constitutive relationship of rock, which is a mathematical formula that describes the mechanical characteristics of rock [1–3]. Geotechnical engineering places a high priority on predicting and understanding rock mass stability. Studying the fundamental relationships of rocks in various contexts is essential due to the complexity and instability of rocks and their environment.

The geomechanical behavior of corroded rock can be accurately reflected by the statistical damage model of rock developed by pertinent researchers using certain mathematical expressions. For example, Xu et al. [4] obtained the chemical damage constitutive model by combining the mechanical damage under an external load with the chemical damage, and verified that the constitutive model can accurately predict the stress–strain relationship after water–rock interaction through a semi-immersion experiment and uniaxial testing. Liang et al. [5] created a constitutive model of acid corrosion sandstone damage based on

the Weibull distribution and Drucker–Prager (D-P) criteria by using sandstone subjected to pH = 1 and 3 hydrochloride (HCl) solutions as the research object. Han et al. [6] established and proved the formula for estimating the damage variable, and they quantitatively evaluated the damage of rock microstructure following chemical corrosion. Pan et al. [7] developed a nonlinear statistical damage constitutive model by using the statistical method, and it can accurately represent the mechanical characteristics of rock. An efficient method for examining the progression of rock degradation is a plausible constitutive model [8–10]. The common features of the above rock damage constitutive models are based on assumptions such as the Weibull distribution; following this, an explicit mathematical expression is used to investigate the relationship between rock stress and strain. However, certain mechanical characteristics, such as peak strength, modulus of elasticity, etc., have a substantial impact on the selection of these model parameters [6,11]. The selection of different parameters is difficult and dependent on the researcher's personal preferences because the mechanical behavior of rocks exhibits highly complicated nonlinear aspects [12,13]. Although some researchers have developed methods for quantitatively predicting rock mechanical characteristics in recent years [14–16], the aforementioned issues cannot entirely be avoided.

New methods, for example, artificial intelligence, virtual reality, and big data, are being developed together with network technology. The study of rock mechanics integrating deep learning has become a new method and trend. In recent years, the constitutive model of rock and soil materials has been studied through artificial intelligence techniques successfully. For example, Ghaboussi et al. [17,18] developed a material constitutive model based on neural networks and successfully applied them to concrete, sand, and synthetic materials in the conceptual model study; neural network research studies were carried out successively by Banimahd [19], Goran [20], Penumadu [21] and others, which further improved the rock constitutive relationship study based on intelligent methods. Chen et al. [22] proposed an evolutionary neural network constitutive model reflecting the total stress–strain characteristics of rock under rock chemical corrosion, on the basis of carefully examining the corrosion effects of rocks under various hydration conditions. Yao [23] developed an evolutionary neural network constitutive model based on genetic algorithms and artificial neural networks, taking into account the confining pressure and hydrochemical corrosion effects by carefully assessing the stress state and chemical damage of limestone under an aqueous chemical solution. The aforementioned findings demonstrate that using neural network technology to create a constitutive model of rock can avoid making assumptions and oversimplifications about its mechanical properties, avoid the need for mathematical expressions of constitutive relationships, and offers a novel method with some promise [17–24]. The constitutive relationship model of fillings under uniaxial compression was created by Qi et al. [25] using the random forest model (RF), which was then verified and analyzed using the measured data. The results revealed that the established RF model could accurately predict the constitutive relationships of fillings under various conditions. This further demonstrates how constitutive models of rock materials may be created using various algorithms (such as random forest). In order to study the applicability in establishing constitutive models of rock, it is important to try different algorithms.

In the machine learning method, SVM performs well in adaptability and fault tolerance, has arbitrary approximation and self-learning capabilities for nonlinear functions, and is appropriate for dealing with nonlinear mapping relationships that call for simultaneous consideration of numerous uncertainties [26–29]. Through the relevant research of some scholars, it can be seen that the ABC algorithm is suitable for optimizing the parameters of the SVM model [30–32], which can improve the generalization ability of the model. Based on these considerations, this study establishes the ABC-SVM rock constitutive model based on the experimental data and introduces performance evaluation indicators ($R^2$, RMSE, and MAPE) to evaluate the model performance and examine the viability of intelligent algorithms in predicting the mechanical behavior. Furthermore, statistical

constitutive models are used to calculate the results, as well as compare and analyze the characteristics and applicable conditions of various models. The paper is organized as follows: Section 1 briefly introduces the application of traditional constitutive models under chemical corrosion and the research results of machine learning in the field of rock mechanics. Section 2 details the principles and flow of the ABC-SVM algorithm. In Section 3, the ABC-SVM constitutive model and the statistical damage constitutive model are established, and the calculation results are compared. Section 4 discusses the advantages and disadvantages of both models, respectively. Finally, the conclusions are provided in Section 5.

## 2. Methods

### 2.1. SVM Algorithm

The SVM algorithm can be used to linearly or nonlinearly separate data, which is a binary classification model. It is a supervised learning method for classification and regression in the field of machine learning [33]. The fundamental goal of the SVM algorithm is to create a hyperplane that can maximally divide the two classes of data while also maximizing the distance (or margin) between the two classes' nearest points (support vectors) and the hyperplane. The kernel function in the SVM algorithm can map indivisible data from low-dimensional space to high-dimensional space, allowing for their separability [34]. Convex quadratic programming can be used to solve the SVM algorithm and find the best hyperplane and kernel function parameters. The SVM model is more suitable for linear regression, and the regression equation can be described as [35]:

$$y_i = \omega \times \phi(x_i) + b \tag{1}$$

where $y_i$ represents the predicted value of sample i, $\omega$ is the normal vector perpendicular to the hyperplane in multidimensional feature space, $x_i$ is a vector consisting of the input variables of the i-th sample, $\phi_i$ is the mapping function, and $b$ refers to the hyperplane vector $\omega$ along the normal.

Through the Lagrange optimization method, optimal constraints are introduced for solving Equation (1):

$$y_i = \sum_{i=1}^{n} (a_i - a_i')\phi(x_i)\phi(x_k) + b \tag{2}$$

where $a_i$ and $a_i'$ are Lagrange multipliers that correspond to the $i$th sample, and $\phi(x_i)\phi(x_k)$ represent the kernel function.

The complex nonlinear problem of rock mechanics is influenced by a wide range of variables. Therefore, it is difficult to utilize linear regression to evaluate its mechanical properties and obtain reliable results. The Kernel function must be used to map the data into a higher-dimensional feature space in order for the SVM model to tackle nonlinear issues. The sigmoid kernel function, radial basis function (RBF) kernel function, and polynomial kernel function are examples of kernel functions. The preferred function among these, the RBF kernel function, is the strongest at dealing with nonlinear problems. In this study, the SVM model is defined using the RBF kernel function. The RBF kernel function can be expressed as [36]:

$$k(x_i, x) = \exp\left(-\frac{||x - x_i||^2}{\sigma^2}\right) \tag{3}$$

In Equation (3), $\sigma$ is the width parameter of the RBF kernel function.

### 2.2. ABC Algorithm

Karaboga et al. [37] created the artificial bee colony optimization method in an effort to mimic the difficult optimization challenge of bee colony search behavior. The main assumption is as follows [30]: Bee scouts look for abundant food sources initially. They are referred to as observation bees when they are looking for food sources. After performing a "swing dance" to help other bees identify the best food source, observation bees will

select a source of honey based on information from other bees and the likelihood that the source of honey will be found. They will then use a search procedure similar to other bees to re-search the field. The observation bees are currently referred to when the word "bees" is used. The outcomes of this cyclic search get closer and closer to the optimal solution, and the related search neighborhood gradually shrinks.

Considering the initial position of the honey source, the initial solution can be solved with Equation (4) [38]:

$$X_{ij} = X_j^{\min} + \text{rand}(0,1)\left(X_j^{\max} - X_j^{\min}\right) \tag{4}$$

in which i = 1, 2, …, N, j = 1, 2, …, D, N is the number of honey sources and D is the population count.

For each solution, a new solution of the adjacent solution $X_k$ of $V_{jk}$ is:

$$V_{jk}(i+1) = X_{jk}(t) + \varphi_{jk}(t)\left(X_{jk}(t) - X_{wk}(t)\right) \tag{5}$$

$$K = \text{int}(\text{rand} \times N) + 1 \tag{6}$$

in which the value range of $\varphi_{jk}$ is between $-1$ and 1, and $X_{jk}$ is the j-th solution of the k-th population solution.

The old response will be replaced by the new one if it is more appropriate. The scout bee selects one answer based on each one's likelihood and fitness value (or mistake). The observation bee will next look for a different response; if it is more pertinent than the original, it will be selected. The ideal choice is made using the fitness value or mistake. When the error is less than the set termination error, the search is finished. To avoid falling into the trap of a local optimal solution, a honey source is calibrated as a tabu search point when the number of iterations (number of cycles) reaches the predetermined limit value and has not been increased. The formula for calculating the probability of observation bees choosing a honey source is as follows:

$$P_i = \frac{fit_i}{\sum_j^N fit_j} \tag{7}$$

According to the basic principle of artificial bee colony algorithm, the main parameters involved in the algorithm include the number of honey sources, termination error, limit, maximum number of iterations, etc. These parameters need to be set according to specific problems when the algorithm is applied.

*2.3. ABC-SVM*

The performance of SVM when used for regression analysis is influenced by the kernel function g and the penalty factor c. The penalty factor c is mostly utilized to lessen prediction error and balance the algorithm's complexity. SVM's capacity for generalization is enhanced by reducing the empirical risk function. The number of support vectors depends on the size of the g value, which, in turn, impacts the generalizability of the model. The generalizability of SVM is typically inversely proportional to the g value. The model's training time is too long when the g parameter is too little, though. In order to obtain the prediction error and generalizability of SVM in the proper state, a suitable g value must be chosen. Therefore, it is crucial to understand how to choose the ideal values for c and g (hyperparameters). The SVM hyperparameters c and g can be improved using the grid search method. Choosing the search range based on existing information or suggested values, then dividing it into grids of a certain length, is the basic notion underlying the grid search strategy. The data (training set) in each group (c, g) were subjected to regression analysis using SVM, and the predicted outcomes were recorded. A value must be sought at each node of the grid. Following a thorough examination of all node values in the grid and

a comparison and analysis of the SVM's prediction results under multiple (c, g) conditions, the (c, g) combination with the lowest error value was identified. The grid search approach can generally search for a better parameter combination and is convenient and quick; however, it is easy to slip into the local minimum trap. This paper uses ABC to optimize SVM, thereby avoiding the local minimum trap in the process of parameter optimization and improving prediction performance and generalizability, to establish a prediction model of rock mechanical properties with better performance and stronger generalizability.

The specific steps of the ABC-SVM algorithm are as follows:

(1)　The data set is established and randomly divided into a training set and test set.

(2)　The parameters of the ABC algorithm are initialized and produce the initial solution.

(3)　The SVM with the initial solution as the parameter combination is used to establish the model on the training set and back-judge to obtain the results. At the same time, the K-fold cross-validation method (K = 5 in this paper) is used to calculate the performance of the model (the mean square error is used as the evaluation index in this paper).

(4)　Honey bees search for new nectar sources using the neighbor search method. Repeat step 3 and compare it with the original outcome, to compare the performance of SVM under different combinations of two parameters. The combination of SVM parameters with the better performance will be retained.

(5)　The observed bee takes the position of the retained nectar source, i.e., (c, g) combination, as the new initial solution, and then repeats step 4.

(6)　Repeat the above steps and record the global optimal solution and the corresponding performance index.

(7)　When the bee passes through the limit cycle, it is judged whether the condition is satisfied. If it is satisfied, the new solution is used instead of the old solution.

(8)　Determine whether the termination condition is satisfied. If it is satisfied, the optimal solution of the output is used as the optimal parameter combination; if it is not satisfied, turn to step 4 until the end condition is satisfied.

(9)　The optimal (c, g) combination obtained by the ABC algorithm is brought into SVM to establish the ABC-SVM model, which is applied to the test set to analyze its performance and generalization ability.

The steps of the 5-fold cross-validation method are as follows:

(1)　The training set is randomly divided into five subsets with the same number of samples in each; four of those are selected as the sub-training set, and the remaining subset is the validation set.

(2)　Based on the initial parameters and sub-training sets of the ABC algorithm, the model is established and then applied to the validation set, and the performance of the model on the validation set is calculated.

(3)　Step 2 is repeated five times, in which the validation set is changed in each cycle to ensure that each set of samples in the training set can be used to train and validate the model.

(4)　The results of the above five cycles are recorded and the average value is calculated.

(5)　Based on the new set of parameters of the bee colony near search, repeat steps 2–4.

(6)　Carry out steps 4–5 of the artificial bee colony algorithm, in which the new parameter performance calculation in step 5 repeats steps 1–5 of the 5-fold cross-validation, and then move on to steps 6–9 of the artificial bee colony algorithm.

The process of predicting rock mechanical properties using the ABC-SVM algorithm is shown in Figure 1.

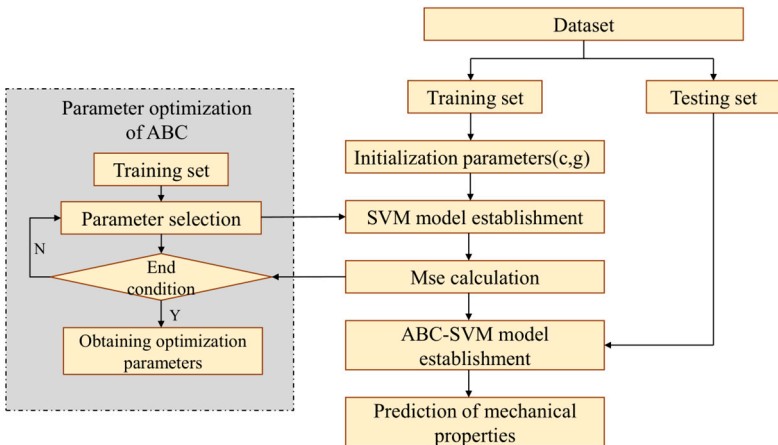

**Figure 1.** ABC-SVM algorithm flow chart.

### 3. Rock Constitutive Model Based on ABC-SVM Model

*3.1. Data Analysis*

The selection of input variables is essential to obtaining calculations with more precise outcomes. In general, the following guidelines [39] must be followed while choosing the input variables: (1) the physical meaning of the parameters is obvious; (2) finding the parameter values is easy; and (3) the parameters can accurately reflect the traits of the output variables. The stress–strain connection of rock is primarily described by the rock constitutive model, where strain is the input variable and stress is the output variable. The mechanical behavior of rock will be influenced by hydrochemical corrosion, confining pressure, and other elements. There may be pores in rock that has been impacted by hydrochemical processes. In direct proportion to the porosity following corrosion, chemical damage to rocks and the severity of mechanical property attenuation both rise. The strength, elastic modulus, peak strain, and ductility of rock all rise with an increase in confining pressure. Thus, confining pressure and porosity are chosen as the input variables. At the same time, the peak point must also be determined because the rock's peak strength will also affect its stress–strain curve. Only the uniaxial compressive strength of the rock can be used to calculate the peak strength of the rock under various confining pressures once the porosity and confining pressure have been established. Therefore, the uniaxial compressive strength of rock is selected as the input variable.

The ABC-SVM rock constitutive model's input variables are porosity, confining pressure, uniaxial compressive strength and strain. The model's output variable is the stress that corresponds to the strain. As a result, the implicit formulation of the ABC-SVM-based rock constitutive relation is [38]:

$$\sigma_1 = f(n, \sigma_3, \sigma_c, \varepsilon) \tag{8}$$

In Equation (8), $\sigma_1$ represents the principal stress of rock under triaxial compression, $n$ denotes porosity, $\sigma_3$ is the confining pressure, $\sigma_c$ is the uniaxial compressive strength, and $\varepsilon$ represents the strain.

As a way to evaluate the effectiveness of the five approaches or models, the correlation coefficient ($R^2$), root mean square error (RMSE), and mean absolute percentage error (MAPE) are introduced in this study. The correlation coefficient, which normally ranges from 0 to 1, measures the strength of the association between the predicted value and the test value. The higher the value, the better. The mean absolute percentage error is the average value of the absolute percentage error, and the discrepancy between the predicted value and the test result is primarily depicted by the root mean square error. The degree of deviation from the observed value is predicted by the two main response models, and the lower the value, the better. The three indicators can be obtained directly from the literature, so the calculation formulas and procedures are not described in detail.

With the use of data on rock porosity, confining pressure, uniaxial compressive strength, and rock mechanics test results, the data set for the ABC-SVM constitutive model was created in this study. By using the research of Qi et al. [25] as a guide, the stress and strain data from the experiment with an interval of 0.1% were chosen in order to shorten the operating time. The data set used in this study was compiled from 60 test samples of sandstone that underwent nuclear magnetic resonance and mechanical testing under various hydration conditions. The statistical box plots of the input variables are displayed in Figure 2. The figure demonstrates that all variables were asymmetrical since their median values did not fall within the box plot's center. In addition, none of the variables had out-of-the-ordinary values, suggesting that the test data could be utilized to build a data set for the prediction of rock mechanical properties. Stratified sampling was carried out based on varied confining pressures and hydration levels. As indicated in Table 1, four rock sample data were chosen as the test set, while further sample data served as the training set. The 5-fold cross validation method was used to select the parameters, to reduce the influence of random error in the process of model training.

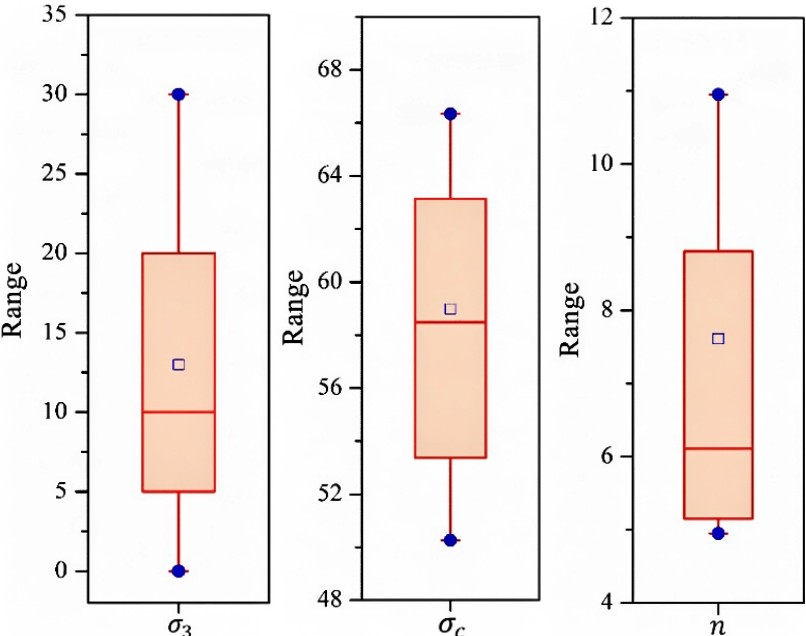

**Figure 2.** Box plot of input variables.

**Table 1.** Testing samples.

| Hydrating Condition | Confining Pressure /MPa | Porosity /% | Uniaxial Compressive Strength/MPa |
|---|---|---|---|
| $H_2SO_4$ | 20 | 10.86 | 52.45 |
| Distilled water | 5 | 5.96 | 63.13 |
| NaOH | 10 | 8.57 | 55.74 |
| Natural state | 0 | 5.02 | 65.16 |

*3.2. Establishment and Verification of Model*

The ABC-SVM model's parameter optimization is shown in Figure 3. The image demonstrates how the ABC-SVM model's mean square error decreases as the number of iterations rises, from 0.055 at the start to 0.0016, and stays nearly unchanged after 20 iterations. The data presented above show how well the ABC algorithm optimizes SVM parameters.

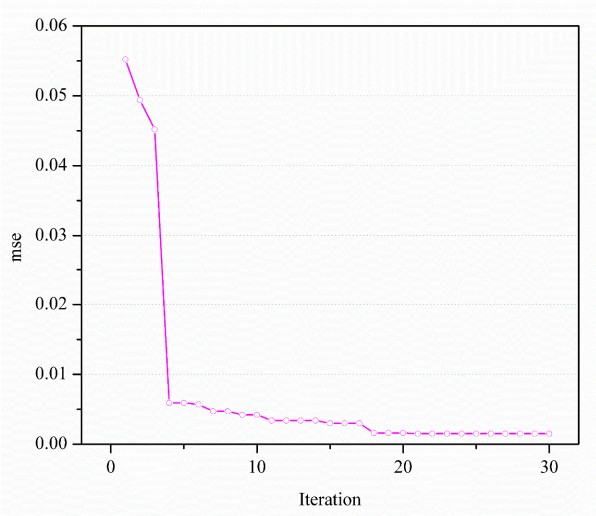

**Figure 3.** The iterative process of ABC-SVM model parameter optimization.

The calculated ideal parameters of the ABC-SVM model are c = 256 and g = 1.7028. On a few samples from the training set, Figure 4 contrasts the test results with the back-judgment outcomes of the optimized ABC-SVM model. The figure shows that the back-judgment results of the model agree with the results of the test. Due to the rock's apparent brittleness and the uniaxial compression's quick drop of post-peak stress, there is a large difference between the expected outcomes of individual locations and the actual values. The test curve itself is also not smooth. Overall, however, the rock's stress–strain relationship may be calculated with accuracy using the constitutive model based on ABC-SVM.

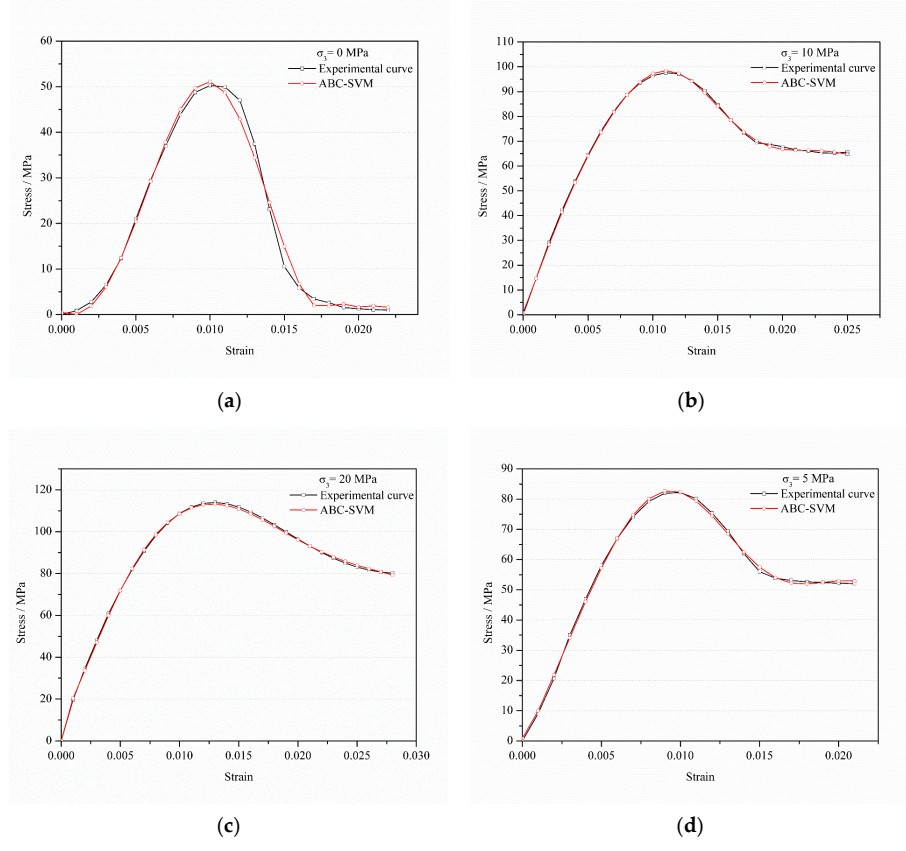

**Figure 4.** Comparison of model back-judgment results with experimental results. (**a**) $H_2SO_4$, $\sigma_3 = 0$ MPa; (**b**) distilled water, $\sigma_3 = 10$ MPa; (**c**) NaOH, $\sigma_3 = 20$ MPa; (**d**) natural state, $\sigma_3 = 5$ MPa.

The back-judgment outcomes of the ABC-SVM constitutive model on the training set are assessed using the correlation between the prediction results and the actual values, as illustrated in Figure 5. The data points are shown in the figure to have a good linear distribution throughout the entire range and to be symmetrically distributed around the best-fit line. The correlation coefficient between the predicted results and the actual value is 0.994, indicating that the back-judgment results of the ABC-SVM constitutive model after parameter optimization are extremely accurate.

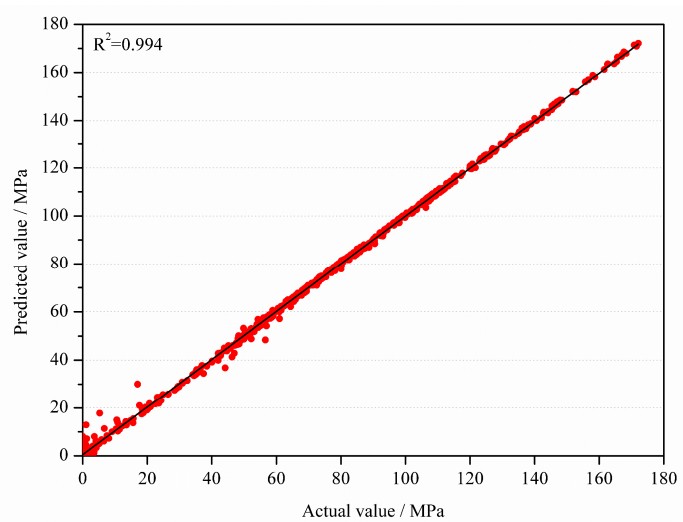

**Figure 5.** The back-judgment results of ABC-SVM model on the training set.

The optimized model is used to calculate the constitutive relationship of sandstone samples in the test set in order to further confirm the predicting capability of the ABC-SVM model on untrained rock samples. A comparison of the experimental findings of the model on the samples from the test set is shown in Figure 6. The calculated curve of the model is almost exactly in line with the test curve, showing that the model can simulate the rock stress–strain relationship very accurately, with the exception of the large error of individual points (mainly in the yield stage and post-peak stage). The stress–strain curve of rock exhibits notable nonlinear characteristics, particularly in the yield stage and post-peak strain softening stage, and the mechanical behavior and characteristics of rock are highly condition-dependent, whereas the sample test data in the training set are insufficient to completely and accurately reflect the mechanical properties of the samples in the test set. Even so, the basic tendency of the test curve might still be visible in the anticipated results. It is evident that the ABC-SVM model has a strong nonlinear prediction capacity and can accurately simulate the triaxial mechanical behavior of rock under different hydration conditions. It might provide a novel viewpoint in the significant area of study known as rock constitutive connection research.

*3.3. Comparison with the Statistical Damage Constitutive Model of Sandstone*

3.3.1. Establishment of Statistical Damage Constitutive Model for Sandstone

The Lemaitre strain equivalence principle is used to establish the fundamental rock constitutive relationship [40]. Based on damage mechanics theory, the mechanical properties of rock under various conditions (such as after water chemical corrosion, temperature action, freeze–thaw rock, etc.) are then taken into account, and a rock constitutive model that is suitable for use is then determined. According to the findings of other literature [4,5,7], the damage constitutive model of sandstone under chemical and mechanical coupling is as follows:

$$\sigma_1 = (1 - D_c)E_0\varepsilon_1[1 - \lambda + \lambda e^{-\left(\frac{F}{F_0}\right)^m}] + \mu(\sigma_2 + \sigma_3) \tag{9}$$

where $\sigma_1, \sigma_2, \sigma_3$ represent the principal stress in three directions, respectively; $E_0$ stands for the elastic modulus of the benchmark damage state; $\varepsilon_1$ corresponds to the strain of $\sigma_1$; $D_c$ is the chemical damage variable; $\lambda$ is the modification coefficient of the damage variable; and $\mu$ represents Poisson's ratio. Strength parameters of the rock micro-element $F$ and the parameter values for the sandstone constitutive model $m$ and $F_0$ were calculated by Liang [5] and brought into Equation (9).

The theoretical curve of the rock was obtained and compared with the test curve. The sandstone samples under the natural state of confining pressure of 5 MPa and the treated distilled water were used as examples. Figure 7 displays the test curve and calculation curve for the constitutive model. Overall, there is good agreement between the theoretical and experimental curves, mainly reflected in the front of the peak.

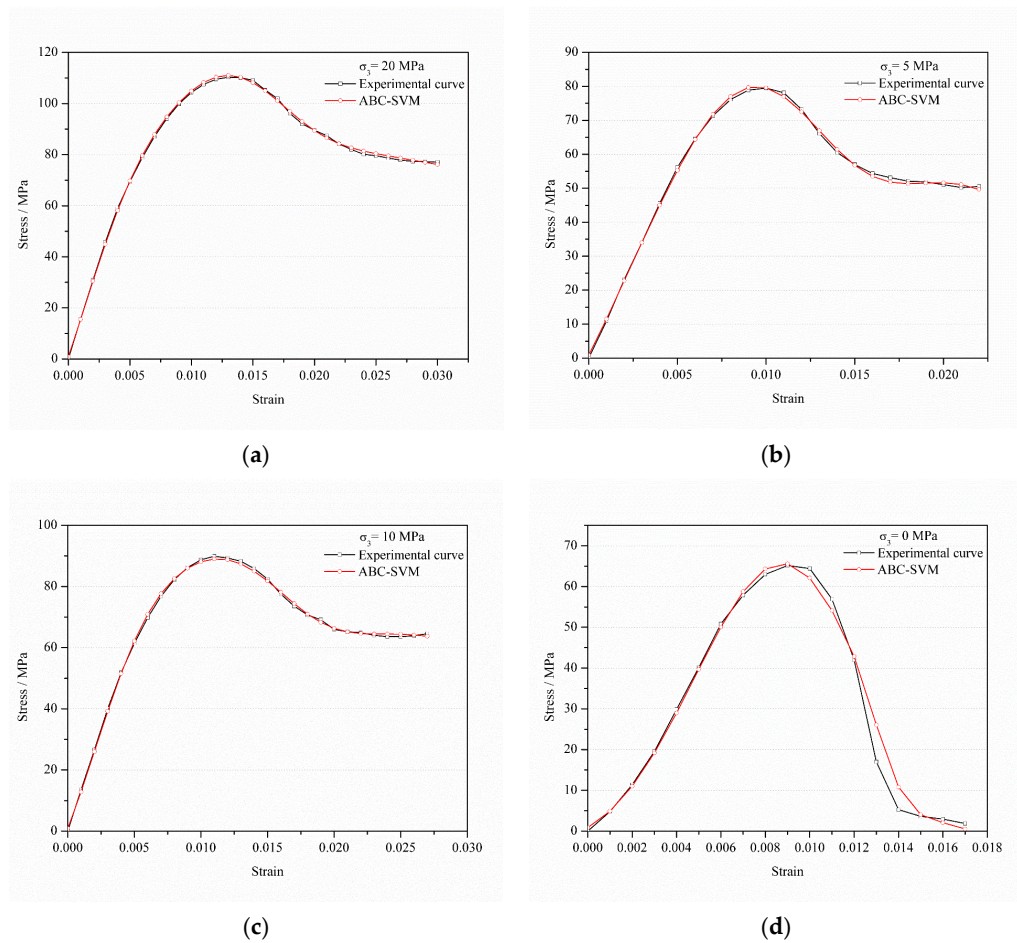

**Figure 6.** Comparison of prediction results and trial results. (**a**) $H_2SO_4$, $\sigma_3$ = 20 MPa; (**b**) distilled water, $\sigma_3$ = 5 MPa; (**c**) NaOH, $\sigma_3$ = 10 MPa; (**d**) natural state, $\sigma_3$ = 0 MPa.

3.3.2. Comparison

In order to quantitatively analyze and compare the performance of the two models, this paper also takes the sandstone samples in the natural state and treated with distilled water as examples under a confining pressure of 5 MPa. The ABC-SVM constitutive model is then used to predict the stress–strain curves of the samples and to compare them with the results of the experiments. Finally, $R^2$, RMSE and MAPE are calculated and analyzed, which are shown in Table 2, and the correlation with the experimental results is shown in Figure 8.

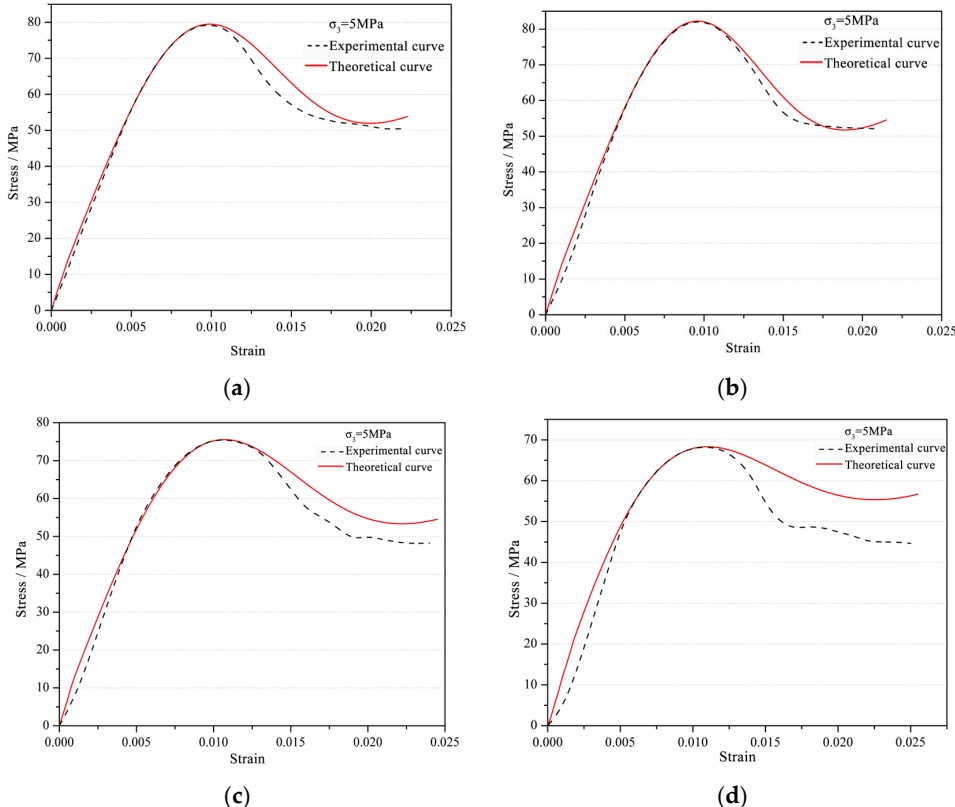

**Figure 7.** Stress–strain experimental curve and theoretical curve of sandstone under different water chemistry conditions (5 MPa). (**a**) Natural state; (**b**) distilled water; (**c**) NaOH; (**d**) $H_2SO_4$.

**Table 2.** The performance indexes of the two models.

| Model | $R^2$ | RMSE | MAPE |
|---|---|---|---|
| Statistical damage constitutive model | 0.990 | 2.5822 | 4.96 |
| ABC-SVM model | 0.998 | 0.7730 | 1.51 |

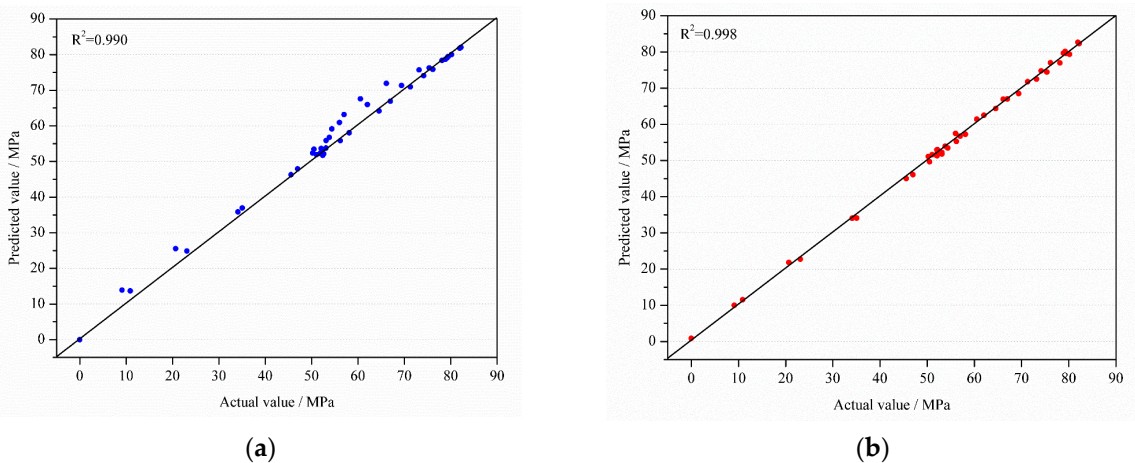

**Figure 8.** The correlation between the simulation results of the two models on the prediction set and the actual values. (**a**) Statistical damage constitutive model; (**b**) ABC-SVM model.

Overall, the data are evenly distributed across the entire range, and there is a general linear relationship between the data points. Additionally, they are evenly placed all around

the best-fitting line. All of the correlation coefficients are greater than 0.9, indicating that both models can produce predictions that are more precise. A comparison of the data points obtained from the test results and prediction results of the rock statistical damage constitutive model reveals that the test results are more off the diagonal than the prediction results of the ABC-SVM model, and the correlation coefficient is also lower. These findings demonstrate that the statistical damage constitutive model and ABC-SVM model developed in this paper can be used to simulate the triaxial compression mechanics experiment of rock under various hydration conditions, and that the calculation outcomes of the ABC-SVM model are more closely correlated with the experimental outcomes.

The ABC-SVM model's error of prediction outcomes is smaller than that of the statistical damage constitutive model, and the results are closer to the experimental values, according to a comparison of the RMSE and MAPE of the two models. A thorough analysis of the two models' performance indicators shows that the ABC-SVM model performs better than the statistical damage constitutive model and is thus better adapted to predicting the nonlinear mechanical behavior of rock under complex conditions. In the case of insufficient data, the statistical damage constitutive model is still recommended for determining the stress–strain relationship of rock.

## 4. Discussion

(1) In rock mechanics, the statistical damage constitutive model of rock is widely used and accepted. Each parameter in the model has a definite physical meaning, which can depict how damage develops during the entire process of rock failure under stress. The number of samples is not limited, and the calculations' findings agree with those of the experiment.

The model has a lot of parameters, and some of those parameters' values are influenced by subjective factors, making it challenging to determine their values with accuracy and increasing the error in the calculation's outcomes. Additionally, the statistical damage constitutive model is typically based on microscopic damage, and the strength distribution features of rock micro-units represent the macroscopic mechanical properties of rock. This also results in a significant discrepancy between the post-peak residual strength section and the test findings since it does not apply to the expansion of macroscopic cracks in the post-peak stage of rock.

(2) The ABC-SVM rock constitutive model's main function is to arbitrarily approximate and learn the nonlinear constitutive relationship of rock in the training set. During the model establishment process, precise mathematical expressions are not necessary, allowing the model calculation outputs to be more closely matched to the test findings while also minimizing errors brought on by incorrect parameter selection. When the size of training samples is sufficiently large and representative, the ABC-SVM model may reflect the constitutive relationship of rock under diverse test settings and be used to predict the constitutive connection of rock in unidentified test situations. However, there are still certain issues with the ABC-SVM rock constitutive model that need to be resolved. For instance, the number of samples in the training set has a significant impact on the model's performance. The model's prediction outcomes will be more precise when there are sufficient representative samples to train it.

## 5. Conclusions

The ABC-SVM rock constitutive model is primarily examined in this research under various hydration circumstances. The ABC-SVM rock constitutive model was created using the experimental data. The model's parameters were tuned using the training set, and the model's performance on the test set after utilizing the optimized parameters was then confirmed. The following are the primary conclusions:

(1) The ABC-SVM rock constitutive model fully takes into account the effects of confining pressure, porosity (chemical deterioration), and other variables. It is appropriate

for modeling the triaxial mechanical behavior of rock under various hydration conditions and has high nonlinear prediction ability.

(2) The ABC-SVM model has a better prediction performance on the typical triaxial constitutive relationship of rock, as evidenced by the larger correlation coefficient ($R^2$ = 0.998) between calculation results and test results and the smaller error (RMSE = 0.7730 and MAPE = 1.51). It is a brand-new approach to describing the composition of rock that has a lot of promise. However, the statistical damage constitutive model is advised when there are few data.

**Author Contributions:** Y.L. and C.Z. conceived and designed the experiments; Y.L. and C.L. performed the experiments; K.Z. and Z.G. contributed materials and theoretical foundations; Y.L. and C.L. wrote the paper. All authors have read and agreed to the published version of the manuscript.

**Funding:** The authors are grateful for the financial support from the National Natural Science Foundation of China (No. 52104109); the Open Fund Research Project, supported by State Key Laboratory of Strata Intelligent Control and Green Mining, co-founded by Shandong Province and the Ministry of Science and Technology (No. SICGM202201); the Natural Science Foundation of Hunan Province of China (No. 2022JJ0602); and the State Key Laboratory for GeoMechanics and Deep Underground Engineering, China University of Mining & Technology (No. SKLGDUEK2209).

**Institutional Review Board Statement:** Not applicable.

**Informed Consent Statement:** Not applicable.

**Data Availability Statement:** Not applicable.

**Conflicts of Interest:** The authors declare no conflict of interest.

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
