# Peer review of "A Constitutive Model Study of Chemical Corrosion Sandstone Based on Support Vector Machine and Artificial Bee Colony Algorithm"

_sustainability, doi:10.3390/su151813415_

Round 1

Reviewer 1 Report

Overall, the paper presents a promising approach to accurately characterize the mechanical properties of rock and its response to hydrochemical corrosion, utilizing a combination of machine learning techniques and experimental data. The manuscript can be accepted with minor revisions.

The work is interesting and clearly written, although it would benefit from a more explicit emphasis on its novelty. Furthermore, I noticed some minor technical errors in the writing. For instance, not all abbreviations are adequately explained (line 37).

The main question addressed by the research is how hydrochemical corrosion affects the mechanical characteristics of rock. The study presents a rock constitutive model based on the artificial bee colony algorithm and support vector machine for accurately predicting rock behavior under different conditions.

I suggest that the authors avoid using abbreviations without explanations in the paper's title and introduction. The topic is original and relevant. I recommend that the authors supplement the obtained results with additional comparisons to other models or chemical simulation tests.

The authors need to provide a more precise description of the novelty and advancements compared to previously published works and the previous use of the method for modeling chemical damage of sandstone.

The conclusion should be supplemented with significant results that support the findings (e.g., RMSE and MAPE).

The references are relevant to the text, but I suggest using more recent publications from recent years.

Figures 2-7 should be clearer, with higher image quality.

Reviewer 2 Report

The article presented research on developing a support vector machine (SVM) rock constitutive model based on the artificial bee colony algorithm (ABC) optimization. The impact of porosity (chemical deterioration), confining pressure, and other aspects are thoroughly studied. This is interesting work but I can't see a good match for this article to be published in Sustainability. In general, while reading the article I didn't see the word sustainable in the whole manuscript. The introduction part must be added with the newest relevant articles. I can see long paragraphs with just one citation in your introduction part. The images/figures must be revised with better resolution.

Overall, the research work's statistical models are in considerable agreement and the work can be considered for publication after minor corrections if the editor agrees that the work matches the journal's aims. 

Minor corrections are needed

Reviewer 3 Report

Clarity and Cohesion: The article's structure is a bit fragmented. Consider reorganizing the sections to flow more logically from the introduction to the conclusion, creating a smoother narrative.

Citations: Ensure proper citation of sources and references to support claims and findings made in the article.: Provide more papers about the SVM algorithms and their kernel functions. Readers with a technical background will appreciate deeper related works such as : 

10.21873/cgp.20063

10.1007/s00521-011-0793-1

10.1109/MVA.2015.7153187

10.1109/ICIP.2015.7350834

Visual Aids: Include figures or maps to visually demonstrate the land cover changes over the study period, along with the classification results for better comprehension.

Grammar and Style: Review the article for grammatical errors and maintain a consistent writing style throughout.

By addressing these points, the article can be enhanced to effectively communicate the study's objectives, methodology, findings, and implications to a broader audience.

 Moderate editing of English language required

Reviewer 4 Report

Thank the authors for submitting this paper to the Sustainability journal. The article requires a significant revision before it can be further considered.

Comment (1): In the literature review section, use a past tense verb for describing literature papers.

Comment (2): The authors repeatedly used the “chemical corrosion” term in the paper. Corrosion is also known as an electrochemical phenomenon where electron transfer occurs. In the context of the present paper, the reviewer wonders if corrosion could also occur electrochemically or if it is just a chemical phenomenon. Further clarification is appreciated.

Comment (3): Several equations were introduced throughout section 2 of the paper. Supporting references must be provided.

Comment (4): The authors must compare and contrast their results with relevant literature sources throughout the results and discussion sections. The reviewer does not see such an in-depth analysis of the data.

Comment (5): The list of references is not up-to-date.

Comment (6): Not a sufficient number of references are cited.  

Comment (7): Add the DOI of the cited references to the list of references.

Minor revision is required, mainly for using "the" in the text. 

Round 2

Reviewer 3 Report

Accepted 

Need small editing 

Reviewer 4 Report

Thank the authors for carefully addressing the reviewer's comments. The manuscript is now sufficiently improved and could be accepted for publication.